# Association of Perceived Neighbourhood Walkability with Self-Reported Physical Activity and Body Mass Index in South African Adolescents

**DOI:** 10.3390/ijerph20032449

**Published:** 2023-01-30

**Authors:** Feyisayo A. Wayas, Joanne A. Smith, Estelle V. Lambert, Natalie Guthrie-Dixon, Yves Wasnyo, Sacha West, Tolu Oni, Louise Foley

**Affiliations:** 1Research Centre for Health through Physical Activity, Lifestyle and Sport (HPALS), Division of Physiological Sciences, Department of Human Biology, Faculty of Health Sciences, University of Cape Town, Cape Town 7700, South Africa; 2Caribbean Institute for Health Research, Epidemiology Research Unit, The University of the West Indies, Mona, Kingston 7, Jamaica; 3Health of Populations in Transition (HoPiT), Research Group, Faculty of Medicine and Biomedical Sciences, University of Yaoundé 1, Yaoundé P.O. Box 8046, Cameroon; 4Department of Sport Management, Cape Peninsula University of Technology, Cape Town 7705, South Africa; 5Medical Research Council Epidemiology Unit, University of Cambridge, Cambridge CB2 0QQ, UK; 6Research Initiative for Cities Health and Equity (RICHE), School of Public Health and Family Medicine, University of Cape Town, Cape Town 7925, South Africa

**Keywords:** adolescents, environmental walkability attributes, physical inactivity, obesity, residential income areas, physical activity domains

## Abstract

Adolescence is a life stage critical to the establishment of healthy behaviours, including physical activity (PA). Factors associated with the built environment have been shown to impact PA across the life course. We examined the sociodemographic differences in, and associations between, perceived neighbourhood walkability, PA, and body mass index (BMI) in South African adolescents. We recruited a convenience sample (n = 143; 13–18 years; 65% female) of students from three high schools (middle/high and low-income areas). Participants completed a PA questionnaire and the Neighbourhood Environment Walkability Scale (NEWS)-Africa and anthropometry measurements. Multivariable linear regression was used to examine various relationships. We found that, compared with adolescents living in middle/high income neighbourhoods, those living in low-income neighbourhoods had lower perceived walkability and PA with higher BMI percentiles. The associations between neighbourhood walkability and PA were inconsistent. In the adjusted models, land use diversity and personal safety were associated with club sports participation, street connectivity was positively associated with school sports PA, and more favourable perceived walkability was negatively associated with active transport. Overall, our findings suggest that the perceived walkability of lower income neighbourhoods is worse in comparison with higher income neighbourhoods, though the association with PA and BMI is unclear.

## 1. Introduction

Physical inactivity, a modifiable risk factor of non-communicable diseases (NCDs), overweight and obesity, is increasing globally in all age groups, including adolescents. Studies in 2012 and 2016 reported that at least 80% of adolescents did not meet daily physical activity guidelines of at least 60 min of moderate-to-vigorous physical activity (MVPA) [1,2]. Another international study indicated that overall, 76% of boys and 85% of girls did not meet physical activity guidelines, with little variation between countries [3]. Patterns of physical inactivity in South Africa are consistent with global trends. A 2019 study that assessed physical activity subjectively indicated that over 90% of adolescents in South Africa did not meet physical activity guidelines (albeit using organized sports as a proxy for physical activity) [4]. An earlier 2012 study that subjectively assessed physical activity across all domains in South African children aged 8–14 years indicated that approximately a third (31%) did not meet guidelines [5]. These differences in prevalence are likely due, in part, to methodological differences, and there are few studies in this population that have used validated physical activity questionnaires. In tandem with rising physical inactivity, the global rate of overweight and obesity in children aged 5–19 years has more than quadrupled between 1975 and 2016 [6]. This prevalence has reached alarming levels among all age groups, especially in low- and middle-income countries (LMICs) [1,7,8,9,10]. For adolescents, this trend translates to an increased risk of NCDs such as cardiovascular diseases and diabetes later in the life course.

Previous studies have demonstrated associations between neighbourhood environmental attributes and physical activity in adolescents, especially in urban areas [11,12]. A recent study conducted among American adolescents showed that a more walkable environment was associated with higher levels of physical activity [13]. This suggests that built environment interventions may have potential to impact adolescent physical activity levels. In a 2011 review of studies conducted in Asia, Europe, Australia, New Zealand, and North America on the neighbourhood environment and physical activity among youth, evidence found associations of physical activity with residential density and land-use mix, with inconsistent evidence for parks, recreation facilities, and street connectivity [14]. Additionally, a more recent study conducted among American adolescents showed that a more walkable environment was associated with increased physical activity in adolescents [13]. Consequently, environmental interventions have been endorsed by the World Health Organization to promote physical activity worldwide [10,15].

Investigations on the association between the built environment and physical activity have been mostly conducted in adult populations, and those studies that do include adolescents are based largely in the Global North [16,17,18]. As such, there are limited studies that have been conducted in South Africa on urban adolescents, a country with a high prevalence of obesity and physical inactivity [5,6,19,20], a history of environmental and social injustices, and which remains one of the most inequitable countries in the world.

Understanding adolescents’ perceptions of the supportiveness of their environment for physical activity could lead to interventions to address declining physical activity levels in this age group specifically [21]. However, there may also be important variations by gender and neighbourhood deprivation levels [16,22]. The interpretation of what is a walkable environment for adolescents, and factors that may influence these perceptions, may help to adapt or focus upstream interventions tailored to different age, gender, and area-level income contexts. Therefore, this study aimed to examine sociodemographic differences in perceived neighbourhood walkability, physical activity, and body mass index (BMI) in South African adolescents. We further sought to investigate the association between perceptions of neighbourhood walkability and both physical activity and body mass index (BMI) in South African adolescents. We hypothesized that:(1)There are sociodemographic differences in perceived neighbourhood walkability, physical activity, and body mass index (BMI) in South African adolescents residing in low-income areas when compared to those residing in middle/high-income areas and;(2)The perceptions of neighbourhood walkability attributes are significantly associated with levels of physical activity and body mass index (BMI) in South African adolescents.

## 2. Methods

### 2.1. School Selection

The Cape Town socioeconomic index map (Appendix A) was used to categorize the location of the schools from which the adolescents were recruited. The very needy/needy areas (resource-poor/low socioeconomic areas) were categorized as low-income, while the average and above average socioeconomic areas were categorized as middle/high-income. Purposive selection of schools was conducted with recruitment from three high schools in Cape Town: two in middle/high-income areas and one in a low-income area. Data were collected between July and September 2019.

### 2.2. Participants

A convenience sample of 143 students aged 13–18 years (grade 8–11) were recruited from the three schools. All age-eligible adolescents were invited to participate, and interested participants were given informed assent and parent or caregiver consent forms. Consent forms for the parents or caregivers of the students were provided in any of the three major languages in Cape Town: English, Xhosa, and Afrikaans. Adolescents who returned their signed assent and consent forms were eligible to be part of the study. After recruitment, the income categories of the residential and school addresses of participants were ascertained (as low vs. medium/high income) (obtained from their demographic data) using the Cape Town socioeconomic index map.

### 2.3. Ethical Considerations

This research study was approved by the University of Cape Town Health Sciences Research Ethics Committee (HREC REF 088/2019). Additional approval to conduct the research in high schools in Cape Town was obtained from the Western Cape Department of Education before beginning the study. Permission from the school authorities was also obtained before school participation in the study. Written assent was obtained from all participants, and written parental consent was obtained from their parent or caregiver. Each student received a ZAR50 grocery voucher or something similar to compensate for the time taken to complete the questionnaires.

### 2.4. Measurements

All measurements and questionnaires described below were administered in the selected schools. Details on data collection are provided below and summarized elsewhere (23). Adolescents in schools had pre-arranged appointments before, during, or after school hours as approved by the school authorities to participate in the study.

### 2.5. Anthropometry

Height, weight, and waist circumference were measured using standardized procedures [23] by a team of trained research assistants using a stadiometer, calibrated scale, and nonelastic fiberglass retractable anthropometric tape, respectively. Waist circumference was taken twice, with a third measurement taken if the difference between the first two measurements was more than 0.5 cm apart, and the average of the measurements was obtained. Body mass index (BMI) was calculated as weight in kilograms (kg)/height in (m)^2^, and thereafter, a children’s BMI-for-age percentile calculator that is age- and sex-specific was used [24].

### 2.6. Questionnaires

#### Demographic Variables

The questionnaire included questions about the adolescent’s age, date of birth, gender, grade, and area of residence (home neighbourhood).

### 2.7. Physical Activity Questionnaire

The physical activity questionnaire was adapted from an instrument that has been previously used in South African adolescents and has been validated against objectively measured physical activity [22]. The self-administered questionnaire solicited responses about the adolescents’ physical activity behaviours during a typical/usual week at different intensities (light, moderate, and vigorous) across the following domains: (1) physical activity at school during physical education (PE) classes; (2) informal physical activity (during school breaks or after school) such as riding a bike, playing with a ball or tag/chasing games; (3) school sports; (4) sports team/sports club activities (not through school); (5) active transport (walking and cycling) to and from school; and (6) household chores such as washing dishes and cleaning. Total physical activity in moderate and vigorous intensity were computed in minutes/week for the different domains.

The different levels of physical activity were described and stated in the validated PA questionnaire [22]. Light physical activity was stated as a slight increase in breathing. It requires standing up and moving around either in the home, workplace, or community. Moderate physical activity was described as leading to a noticeable increase in depth of breathing while still allowing for comfortable talking. Vigorous physical activity was described as leading to harder breathing or puffing and panting and not being able to speak more than a few words without pausing for breath. As overestimation is very common with self-reported measurements of physical activity [25,26], for the current study, physical activity durations were truncated to avoid improbable high values in the different domains as follows: school sports and club sports at 1800 min/week; informal physical activity at 1680 min/week; and active transport at 1260 min/week. Additionally, we truncated total physical activity at 2400 min/week [25,26]. Physical activity for PE was not truncated, as this domain was structured in schools. We did not truncate values in the chores domain, as most activities were classified as light physical activity and thus did not contribute to the calculation of total moderate-to-vigorous physical activity.

#### Perceived Neighbourhood Environment Features

Perceptions of the neighbourhood environment were assessed using the Neighbourhood Environment Walkability Survey-Africa (NEWS-Africa). NEWS is a tool that has been validated in sub-Saharan African countries including Kenya, Nigeria, Uganda, and South Africa [27,28,29]. NEWS-Africa assesses different aspects of the built environment, namely: (1) residential density; (2) land-use mix—diversity (proximity to non-residential destinations); (3) land-use mix—access (ease of access to services and places); (4) street/road connectivity; (5) infrastructure and safety for walking; (6) aesthetics; (7) traffic safety; and (8) safety from crime. Additional single items for adolescents similar to a previous study were included to suit the context, namely: (9) personal safety and (10) stranger danger [17].

As such, the NEWS-Africa in our study included 10 subscales. All the subscales were computed as the mean of responses to items in the scale, with responses coded (or reverse coded) such that higher values indicated a more favourable response to the environmental characteristic and therefore demonstrated greater walkability [29]. A general “walkability index” was constructed by computing the mean of their standardized scores of the NEWS-Africa subscales to operationalize the larger construct of walkability [28]. A higher score for the walkability index indicated a more walkable neighbourhood.

## 3. Data Analysis

Analyses were conducted using SPSS 25.0 software (SPSS Inc., Chicago, IL, USA). Mean and standard deviations, or medians and interquartile ranges, were used to generate descriptive statistics for continuous variables. Frequency tables were used to generate descriptive statistics for categorical variables. Mann–Whitney U was used to test for differences in demographic variables, anthropometry, self-reported PA, and NEWS by home socioeconomic area (low vs. middle/high income) and gender (males vs. females). A multivariable linear regression method with a univariate model was used to analyze the relationship between built environment perceptions with physical activity (total and domain-specific, but excluding PE, which is a structured activity in the schools) and BMI. Statistical significance was set at *p* < 0.05.

Multilevel linear regression analyses were conducted to examine the direct associations between NEWS variables as independent variables with domains and total physical activity as dependent variables, adjusting for home and school socioeconomic category and gender as independent covariates. Preliminary analyses were done to ensure the normality, linearity, and multicollinearity were met. As the physical activity outcomes had a skewed distribution, their median and interquartile range values were reported in the descriptive table, and the square root of their original variables was used in the regression analyses to improve their normality. Additionally, the z scores of the NEWS variables were used for the analysis. Before conducting the analyses, multicollinearity between the independent variables was tested with Pearson correlations and the variance inflation factor (VIF). None of the correlation coefficients exceeded r = 0.7 (highest r = 0.60); and the VIF values were below 10. The NEWS walkability index was analyzed as a simple linear regression, as it was computed on the aggregate of all the NEWS variables, which will result in high multicollinearity. All sociodemographic variables and neighbourhood strata were entered as covariates in the first block of the regression models, and environmental variables were added as a second block. We estimated separate covariate-adjusted multilevel linear regression for each environmental attribute (single-environmental attribute models) and all environmental attributes entered in the model simultaneously (multiple-environmental attribute models).

## 4. Results

Overall, the median age was 15 years old with almost two-thirds (65%) of the adolescents being female. Table 1 shows the sample characteristics (demographics, anthropometry, physical activity, and NEWS subscales) categorized by income area. Adolescents from low-income areas had a higher waist circumference and BMI percentile when compared with those from the middle/high-income areas. Overall, about half (51.7%) of the adolescents met the minimum recommended weekly physical activity of 420 min, with no significant difference when categorized by residential income area. Adolescents from middle/high-income neighbourhoods spent more time in school sports and total physical activity, and less time in chores, than adolescents from low-income neighbourhoods. The NEWS analysis indicated for the subscales: land-use mix, land-use diversity, places for walking, cycling, and playing, aesthetics, safety from crime, safety from traffic, and the overall walkability index, adolescents from middle/high-income areas had significantly higher values (indicating more favourable perceptions of walkability) compared with the adolescents from low-income areas.

Table 2 shows the sample characteristics (demographics, anthropometry, PA, and NEWS subscales) categorized by gender. A significantly higher proportion of male adolescents met the recommended PA (68.0% vs. 43.0%, *p* = 0.004) and had lower BMI percentiles (58.5% vs. 70.8%, *p* = 0.005) when compared with the female adolescents. Male adolescents spent more time in club sports, informal activities, and total physical activity as well as less time in chores than female adolescents. There were no differences between males and females across the NEWS subscales, apart from perceptions of land-use diversity, which were more favourable in males than females.

Table 3 presents unadjusted associations of perceived neighbourhood environmental attributes with physical activity domains and total physical activity, and Table 4 presents the adjusted associations. In the maximally adjusted models, land-use diversity and personal safety were associated with club sports, albeit in different directions (land-use diversity ß: 2.15, 95% CI: 0.66; 3.64, *p* = 0.05; personal safety ß: −1.69, 95% CI: −3.18; 0.19, *p* = 0.03). Street connectivity was positively associated with school sports (ß: 1.29, 95% CI: 0.03; 2.56, *p* = 0.04), and more favourable walkability was negatively associated with active transport (ß: 1.53, 95% CI: −2.92; −0.13, *p* = 0.03). Table 5 presents the unadjusted and adjusted associations of perceived neighbourhood environmental attributes with the BMI of the adolescents.

## 5. Discussion

### 5.1. Main Findings and Comparison with Existing Literature

The aim of the study was to examine sociodemographic differences in perceived neighbourhood walkability, physical activity, and body mass index (BMI) in SA adolescents. We further investigated the association between perceptions of neighbourhood walkability and both physical activity and BMI. A better understanding of sociodemographic differences in neighbourhood walkability attributes, physical activity, and BMI is important in co-developing sustainable, effective, and context-specific health behaviour interventions for insufficient levels of PA in adolescents. Even more importantly, this understanding may highlight the upstream and midstream barriers to participation in physical activity, which may require advocacy and policy initiatives. The most important findings of the present study were that, compared with adolescents living in middle/high-income neighbourhoods, those living in low-income neighbourhoods had lower perceived walkability and PA as well as higher BMI percentiles. The associations between neighbourhood walkability and PA were inconsistent. In the adjusted models, land-use diversity and personal safety were associated with club sports participation, street connectivity was positively associated with school sports PA, and more favourable perceived walkability was negatively associated with active transport.

The estimated proportion of adolescents meeting physical activity guidelines by self-report in this study (51.7%) was higher compared to the global finding (less than 20% of children aged 11–17 years) [2] but lower than in a study conducted in South Africa with children aged 8–14 that indicated that 69% of the children met guidelines [5].

We found that low- and middle/high-income neighbourhoods differed fundamentally in terms of perceived supportiveness of the environment for physical activity and, correspondingly, with physical activity and body composition in consistent directions. The scores for most of the perceived walkability subscales were significantly lower in low-income areas. Additionally, adolescents living in low-income neighbourhoods had significantly higher BMI percentiles and waist circumferences as well as lower physical activity levels compared with adolescents living in middle/high-income neighbourhoods. Domains of physical activity differed in ways that may be consistent with socioeconomic/income status. For example, time spent in school sports was higher in the middle/high-income neighbourhoods, whereas time spent in chores was higher in the low-income neighbourhoods. Micklesfield et al. reported similar findings in rural South Africa among adolescents [22]. Socioeconomic differences in body composition likely relate to a range of factors, including diet. Qualitative research in the same sample (not currently published) indicates that adolescents residing in low-income areas reported more barriers to accessing and consuming healthy food options due to financial means, preference of unhealthy options in the household, and lack of food autonomy.

While perceptions of the environment did not differ markedly between males and females, we found gender differences in physical activity (lower in females), BMI percentile and waist circumference (both higher in females). This is consistent with patterns found previously in South Africa and in other countries [4,30]. Females had a higher level of PA from chores and lower levels of MVPA from informal activities and club sports, which is similar to a previous study in urban South Africa [4]. This may be due in part to female adolescents being more involved in household chores. Additionally, parents or guardians may be more aware of the dangers of the neighbourhood than their children are and may restrict outdoor physical activities, especially for female adolescents [21,31]. Female adolescents have been shown to participate significantly less in sports when compared with males in South Africa [32].

The associations between neighbourhood walkability and physical activity were inconsistent. In the adjusted models, more favourable perceptions of land-use diversity and street connectivity were associated with sport participation (club sports and school sports, respectively). Previous studies have indicated that land-use mix (diversity and access) [12,33] is one of the major environmental attributes that is important for adolescents’ physical activity. However, less favourable perception of walkability was associated with more time in active transport, which seems counter intuitive. A South African study indicated that active transport and informal sports were important in meeting the recommended MVPA in adolescents [22]. This indicates the importance of active transport, especially in adolescents from low-income areas when compared to adolescents from middle/higher residential areas. Furthermore, this might be the contributory factor to the negative association of active transport with walkability index, as adolescents, regardless of their environmental attributes, may use active transport to get to school because many of their parents do not own a car [34,35]. As such, active transport seems to be done more by necessity than choice, regardless of the neighbourhood environmental attributes [36,37,38]. A previous study in South Africa indicated that although more than 50% of parents had safety issue concerns, their children were still engaged in active transport due to a lack of other alternatives [38]. In our study, the proportion of adolescents that met physical activity guidelines decreased from 51.7% to 38.5% when active transport was excluded.

We did not find significant associations between built environment perceptions and BMI. Findings from other studies have also been mixed, with some research in the United States of America showing significant association between the built environment and body composition, indicating a relationship [39,40], while other studies found no association [13,41].

### 5.2. Strengths and Limitations

This is one of few studies in South Africa examining the association of perceptions of neighbourhood walkability with physical activity and body composition. Limitations of the study include the cross-sectional design and purposive sampling strategy, which make it prone to research bias, and the inability to generalize to the overall adolescent population in South Africa. In addition, the self-reported measures of both physical activity and perceptions of the environment might have introduced some measurement error and self-report bias.

### 5.3. Recommendations and Policy Implications

We found that land-use diversity and street connectivity were positively associated with engaging in sports in adolescents. These features relate to urban planning, and as such, policy makers should explicitly consider the potential for neighbourhood design to facilitate active lifestyles. Despite the counter-intuitive association found between personal safety and active transport, active transport remained an important contributor to adolescents’ physical activity. As such, transport planning should consider how to facilitate safe and interconnected walking and cycling, including the provision of sidewalks and cycling routes, as well as reducing risks related to road traffic and personal violence. Decision makers in education should consider the important role of school and club sports and ensure that adequate and well-maintained facilities and equipment are available.

## 6. Conclusions

Overall, our findings suggest that the perceived walkability of lower-income neighbourhoods is lower than higher-income neighbourhoods, though the association with physical activity and BMI is unclear. Regardless, this suggests that area level deprivation plays a role in health inequality in this context. Recommended effective interventions to bridge health inequality and equity gaps will therefore need to be tailored to any given context.

## Figures and Tables

**Table 1 ijerph-20-02449-t001:** Overall and specific study condition sample characteristics categorized by residential income area.

Variables	Low Residential Income Area [N = 92]	Middle/High Residential Income Area [N = 51]	Total [N = 143]	*p*-Value
Age	15 (14,16)	15 (14,16)	15 (14,16)	0.34
Sex N (%)				
Male	28.0 (30.4)	22.0 (43.1)	50.0 (35.0)	
Female	64.0 (69.6)	29.0 (59.6)	93.0 (65.0)	0.13
Waist circumference (cm)	69.6 (67.1,76.2)	68.2 (63.3, 73.1)	69.4 (65.2, 75.4)	0.03 *
Body mass index (percentile)	73.0 (51.4, 90.6)	56.7 (23.8, 72.7)	65.7 (43.5, 87.9)	0.01 *
Met physical activity recommendation(yes)N (%)	45.0 (48.9)	29.0(56.9)	74.0 (51.7)	0.36
Domains of physical activity (minutes/week)				
Physical education activities (MVPA)	60.0 (30.0, 60.0)	45.0 (30.0, 60.0)	50.0 (30.0, 60.0)	0.22
Informal activities (MVPA)	60.0 (0.0, 190.5)	72.5 (18.8, 337.5)	60.0 (0.0, 220.0)	0.23
School sports (MVPA)	60.0 (0.0, 120.0)	120.0 (0.0, 242.5)	60.0 (0.0, 150.0)	0.001 *
Club sports (MVPA)	40.0 (0.0, 175.0)	30.0 (0.0, 195.0)	30.0 (0.0, 180.0)	0.67
Active transport (MVPA)	0.0 (0.0, 200.0)	0.0 (0.0, 30.0)	0.0 (0.0, 120.0)	0.07
Chores	657.5 (336.3, 1021.8)	378.0 (117.5, 741.3)	570.0 (250.0, 960.0)	0.001 *
Total MVPA	407.5 (191.3, 695.0)	455.0 (295.0, 840.0)	430.0 (210.0, 765.0)	0.12
Neighbourhood walkability domains				
Residential density	5.0 (2.0, 5.0)	5.0 (3.0, 5.0)	5.0 (3.0, 5.0)	0.39
Access to services and places (land-use mix)	2.9 (2.4, 3.3)	3.1 (2.7, 3.4)	3.0 (2.6, 3.4)	0.02 *
Land-use diversity	2.6 (2.2, 2.8)	3.0 (2.6, 3.5)	2.7 (2.3, 3.0)	<0.001 *
Street connectivity	3.0 (2.1, 3.2)	3.0 (2.4, 3.2)	3.0 (2.2, 3.2)	0.30
Places for walking, cycling, and playing	2.5 (2.2, 2.8)	2.9 (2.7, 3.2)	2.7 (2.3, 3.0)	<0.001 *
Aesthetics	2.4 (1.9, 2.8)	3.0 (2.4, 3.5)	2.5 (2.1, 3.0)	<0.001 *
Safety from crime	2.3 (1.8, 3.0)	3.3 (2.5, 3.8)	2.8 (2.0, 3.5)	<0.001 *
Safety from traffic	2.7 (2.0, 3.0)	2.8 (2.3, 3.3)	2.7 (2.0, 3.2)	0.04 *
Personal safety	3.0 (2.7, 3.3)	3.3 (2.7, 3.7)	3.0 (2.7, 3.3)	0.19
Stranger danger	3.0 (2.3, 3.9)	3.0 (2.7, 4.0)	3.0 (2.7, 4.0)	0.26
Walkability index	−0.1 (−0.6, 0.2)	0.3 (−0.0, 0.6)	0.0 (−0.3, 0.4)	<0.001 *

**Categorical variables are presented as N (%).** All skewed data are reported as median (IQR—25–75th percentile). Abbreviations: PA: physical activity; MVPA: moderate-to-vigorous physical activity. * *p*-values <0.05 represent a significant difference between variables. Non-parametric Mann–Whitney U conducted on the skewed data. A higher score for all the subscales of the built environment (walkability domains) indicated a more walkable neighbourhood.

**Table 2 ijerph-20-02449-t002:** Overall and specific study condition sample characteristics categorized by gender.

Variables	Female N = 93	Male N = 50	Total N = 143	*p*-Value
Age (years)	15 (14, 16)	15 (14, 16)	15 (14, 16)	0.74
Waist circumference (cm)	69.4 (64.4, 76.6)	70.5 (67.2, 73.5)	69.4 (65.2, 75.4)	0.56
Body mass index (percentile)	70.8(48.8, 91.7)	58.5 (36.5, 73.3)	65.7 (43.5, 87.9)	0.005 *
Met MVPA recommendation(yes) N (%)	40.0 (43.0)	34.0 (68.0)	74.0 (51.7)	0.004 *
Domains of PA (minutes/week)	Median (IQR—25–75th percentile)			
Physical education activities (MVPA)	50.0 (30.0, 60.0)	50.0 (30.0, 60.0)	50.0 (30.0, 60.0)	0.90
Informal activities (MVPA)	45.0 (0.0, 170.0)	142.5 (60.0, 397.5)	60.0 (0.0, 220.0)	<0.001 *
School sports (MVPA)	60.0 (0.0, 120.0)	75.0 (00.0, 180.0)	60.0 (0.0, 150.0)	0.09
Club sports (MVPA)	0.0 (0.0, 120.0)	110.0 (00.0, 315.0)	30.0 (00.0, 180.0)	0.003 *
Active transport (MVPA)	0.0 (0.0, 150.0)	0.0 (00.0, 26.3)	0.0 (0.0, 120.0)	0.07
Chores and casual work (PA)	625.0 (300.0, 1062.0)	392.0 (196.3, 690.0)	570.0 (250.0, 960.0)	0.03 *
Total MVPA (including active transport)	345.0 (167.5, 677.5)	567.5 (348.8, 843.8)	430.0 (210.0, 765.0)	0.001 *
Neighbourhood walkability domains				
Residential density	3.0 (1.0, 5.0)	3.8 (1.0, 5.0)	5.0 (3.0, 5.0)	0.45
Access to services and places (land-use mix)	2.9 (2.6, 3.4)	3.0 (2.5, 3.3)	3.0 (2.6, 3.4)	0.98
Land-use diversity	2.6 (2.2, 3.0)	2.8 (2.6, 3.2)	2.7 (2.3, 3.0)	0.01 *
Street connectivity	3.0 (2.3, 3.2)	2.8 (2.2, 3.3)	3.0 (2.2, 3.2)	0.64
Places for walking, cycling, and playing	2.7 (2.3, 3.9)	2.6 (2.3, 2.9)	2.7 (2.3, 3.0)	0.72
Aesthetics	2.5 (2.1, 3.1)	2.5 (2.2, 3.0)	2.5 (2.1, 3.0)	0.83
Safety from crime	2.8 (2.0, 3.5)	2.5 (2.0, 2.5)	2.8 (2.0, 3.5)	0.36
Safety from traffic	2.8 (2.0, 3.5)	2.7 (2.1, 3.0)	2.7 (2.0, 3.2)	0.79
Personal safety	3.0 (2.7, 3.3)	3.0 (2.6, 3.3)	3.0 (2.7, 3.3)	0.30
Stranger danger	3.0 (2.3, 4.0)	3.0 (2.7, 4.0)	3.0 (2.7, 4.0)	0.92
Walkability index	0.01 (−0.3, 0.4)	0.08 (0. 3, 0.5)	0.0 (−0.3, 0.4)	0.50

Categorical variables are presented as N (%). All skewed data are reported as median (IQR—25–75th percentile). Abbreviations: PA: physical activity; MVPA: moderate-to-vigorous physical activity. * *p*-values <0.05 represent a significant difference between variables. Non-parametric Mann–Whitney U were conducted on the skewed data. A higher score for all the subscales of the built environment (walkability domains) indicated a more walkable neighbourhood.

**Table 3 ijerph-20-02449-t003:** Unadjusted associations of perceived neighbourhood environmental attributes with physical activity domains in urban South Africa.

	Total MVPA (ß, 95% CI)	*p*-Value	Active Transport (ß, 95% CI)	*p*-Value	Informal Activities (ß, 95% CI)	*p*-Value	School Sport MVPA (ß, 95% CI)	*p*-Value	Club Sport (ß, 95% CI)	*p*-Value
Neighbourhood walkability domains										
Residential density	−0.50 (−2.28, 1.26)	0.57	−1.36 (−2.72, −0.00)	0.05	−0.24 (−1.94, 1.46)	0.78	0.12 (−1.24, 1.46)	0.87	1.24 (−1.04, 2.58)	0.70
Access to services and places (land-use mix)	0.56 (−0.42, 3.19)	0.54	0.52 (−0.88, 1.92)	0.46	0.74 (−1.01, 2.49)	0.41	−0.34 (−1.73, 1.04)	0.63	−1.05 (−2.43, 0.33)	0.14
Land-use diversity	1.39 (0.54, 3.60)	0.13	−0.88 (−2.28, 0.52)	0.21	0.89 (−0.86, 2.63)	0.32	0.89 (−0.50, 2.27)	0.21	2.31 (0.93, 3.69)	0.001 *
Street connectivity	0.54 (−1.11, 2.20)	0.52	−0.77 (−2.09, 0.51)	0.24	0.68 (−0.92, 2.28)	0.40	1.56 (0.29, 2.83)	0.02 *	−0.39 (−1.65, 0.88)	0.55
Places for walking, cycling, and playing	−0.67 (−2.65, 1.32)	0.51	0.21 (−1.33, 1.74)	0.79	−1.63 (−3.56, 0.29)	0.95	0.46 (−1.06, 1.99)	0.55	0.58 (−0.94, 2.09)	0.45
Aesthetics	1.02 (−1.23, 3.27)	0.37	−1.17 (−2.91, 0.58)	0.19	0.88 (−1.30, 3.06)	0.43	1.40 (−0.33, 3.13)	0.11	0.71 (1.01, 2.43)	0.41
Safety from crime	−1.33 (−3.67, 1.02)	0.63	−0.97 (−2.78, 0.85)	0.29	−0.29 (−2.56, 1.98)	0.80	−0.92 (−2.72, 0.88)	0.32	−0.14 (−1.94, 1.65)	0.87
Safety from traffic	0.02 (−2.20, 2.23)	0.99	0.32 (−1.39, 2.03)	0.71	0.87 (−1.28, 3.01)	0.43	0.06 (−1.65, 1.75)	0.95	−1.33 (−3.02, 0.37)	0.12
Personal safety	−0.39 (−2.32, 1.54)	0.69	−0.71 (−2.21, 0.79)	0.35	0.41 (−1.46, 2.29)	0.66	−0.06 (−1.54,1.43)	0.94	−1.75 (−3.23, −0.28)	0.02 *
Stranger danger	0.87 (−1.36, 3.10)	0.44	0.77 (−0.96, 2.49)	0.38	−1.17(−3.33, 0.99)	0.29	0.76 (−0.95, 2.47)	0.38	1.40 (−0.31, 3.11)	0.11
Walkability index	0.73 (−0.85, 2.32)	0.36	−2.05(−3.29, −0.81)	0.001 *	0.63 (−0.90, 2.16)	0.42	1.83 (0.62, 3.05)	<0.001 *	1.05 (−0.23, 2.33)	0.11

Abbreviations: MVPA: moderate-to-vigorous physical activity; PE: physical education; PA: physical activity; β: regression coefficient; 95% CI: 95% confidence intervals; *p*-value is significant at * *p* < 0.05. A higher score for all the subscales of the built environment (walkability domains) indicated a more walkable neighbourhood.

**Table 4 ijerph-20-02449-t004:** Adjusted associations of perceived neighbourhood environmental attributes with physical activity domains of adolescents in urban South Africa.

	Total MVPA (ß, 95% CI)	*p*- value	Active Transport MVPA (ß, 95% CI)	*p*-Value	Informal MVPA (ß, 95% CI)	*p*-Value	School Sport MVPA (ß, 95% CI)	*p*-Value	Club Sport MVPA (ß, 95% CI)	*p*-Value
**Neighbourhood Walkability Domains**										
Residential density	−0.75 (−2.51, 1.00)	0.82	−1.08 (−2,44, 0.29)	0.12	−0.66 (−2.32, 0.99)	0.43	−0.04 (−1.38, 1.30)	0.95	0.97 (−0.39, 2.32)	0.16
Access to services and places (land-use mix)	0.77 (−1.03, 2.56)	0.47	0.26 (−1.14, 1.65)	0.72	1.10 (−0.59, 2.79)	0.20	−0.13 (−1.50, 1.24)	0.85	−0.87(−2.25, 0.51)	0.22
Land-use diversity	0.40 (−1.53, 2.34)	0.55	−0.17 (−1.67, 1.34)	0.83	−0.39 (−2.22, 1.43)	0.67	−0.02 (−1.49, 1.46)	0.98	2.15 (0.66, 3.64)	**0.05 ***
Street connectivity	0.53 (−1.13, 2.18)	0.31	−0.57 (−1.86, 0.72)	0.38	0.58 (−0.98, 2.13)	0.47	1.29 (0.03, 2.56)	**0.04 ***	−0.38 (−1.65, 0.90)	0.56
Places for walking, cycling, and playing	−0.64 (−2.61, 1.33)	0.47	0.25 (−1.29, 1.78)	0.75	−1.58 (−3.44, 0.27)	0.09	0.21 (−1.29, 1.71)	0.79	0.80 (−0.71, 2.32)	0.30
Safety from crime	−1.19 (−3.56, 1.19)	0.56	−1.11(−2.96, 0.74)	0.24	0.02 (−2.22, 2.25)	0.99	−1.19 (−3.00, 0.62)	0.20	0.39 (−1.45, 2.21)	0.68
Safety from traffic	−0.04 (−2.22, 2.15)	0.92	0.49 (−1.21, 2.19)	0.57	0.71 (−1.35, 2.77)	0.50	−0.05 (−1.72, 1.62)	0.95	−1.47 (−3.15, 0.22)	0.09
Personal safety	0.08 (−1.86, 2.02)	0.42	−1.09 (−2.60, 0.42)	0.16	1.04 (−0.79, 2.86)	0.26	0.43 (−1.05, 1.91)	0.57	−1.69 (−3.18, −0.19)	**0.03 ***
Stranger danger	0.62 (−1.59, 2.83)	0.98	1.00 (−0.72, 2.72)	0.25	−1.58 (−3.66, 0.51)	0.14	0.09 (−0.98, 2.39)	0.41	1.10 (−0.60, 2.80)	0.20
Aesthetics	0.81 (−1.45, 3.06)	0.25	−0.13 (−2.74, 0.77)	0.27	0.66 (−1.47, 2.78)	0.54	0.12 (−0.81, 2.63)	0.30	0.97 (−0.77, 2.71)	0.27
Walkability index	0.29 (−1.47, 2.04)	0.75	−1.53 (−2.92, −0.13)	**0.03 ***	0.18 (−1.52, 1.83)	0.85	0.84 (−0.51, 2.19)	0.22	1.14 (−0.29, 2.56)	0.12

Abbreviations: MVPA: moderate-to-vigorous physical activity; PA: physical activity; β: regression coefficient; 95% CI: 95% confidence intervals. All regression coefficients were adjusted for participant sex and home and school income area. *p*-value is significant at * *p* < 0.05. A higher score for all the subscales of the built environment (walkability domains) indicated a more walkable neighbourhood.

**Table 5 ijerph-20-02449-t005:** Associations of perceived neighbourhood environmental attributes with body mass index of adolescents in urban South Africa.

Variables	BMI Unadjusted (ß, 95% CI)	*p*-Value	BMI Adjusted (ß, 95% CI)	*p*-Value
**Neighbourhood Walkability Subscales**				
Residential density	−0.02 (−0.21, 0.16)	0.81	−0.01 (−1.19, 0.17)	0.89
Access to services and places (land-use mix)	0.07 (−0.12, 0.26)	0.47	0.05 (−0.13, 0.24)	0.57
Land-use diversity	−0.11 (−0.29, 0.08)	0.27	0.04 (−0.15, 0.24)	0.67
Street connectivity	−0.14 (−0.32, 0.03)	0.10	−0.14 (−0.30, 0.03)	0.12
Places for walking, cycling, and playing	−0.09 (−0.30, 0.120)	0.39	−0.07 (−0.27, 0.13)	0.51
Safety from crime	−0.02 (−7.20, 6.14)	0.88	0.02 (−0.22, 0.27)	0.85
Safety from traffic	0.06 (−0.17, 0.29)	0.61	0.05 (−0.17, 0.27)	0.65
Personal safety	0.16 (−0.04, 0.36)	0.12	0.09 (−0.11, 0.29)	0.38
Stranger danger	−0.05 (−0.29, 0.18)	0.65	−0.05 (−0.27, 0.18)	0.67
Aesthetics	−0.12 (−0.35, 0.12)	0.33	−0.05 (−0.28, 0.18)	0.69
Walkability index	−0.25 (−0.57, 0.07)	0.12	0.03 (−1.58, 2.19)	0.75

Abbreviations: BMI: body mass index; β: regression coefficient; 95% CI: 95% confidence intervals. Adjusted for participant sex and home and school income area. *p*-value is significant at *p* < 0.05. A higher score for all the subscales of the built environment (walkability domains) indicated a more walkable neighbourhood.

## Data Availability

The data presented in this study are available on request from the corresponding author.

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
