# Peer review of "Association of Perceived Neighbourhood Walkability with Self-Reported Physical Activity and Body Mass Index in South African Adolescents"

_ijerph, 2023, doi:10.3390/ijerph20032449_

Round 1

Reviewer 1 Report

Overall, it is a good paper. They used a granular approach that adds strength to the methods. However, minor revisions in the grammar and language are suggested. 

Reviewer 2 Report

Manuscript ID:  ijerph-2159501

Association of perceived neighbourhood walkability with self-reported physical activity and body mass index in South African adolescents

The manuscript fits with the aim of the ijerph, and the subject reveals good content for researchers and professionals in the subject of (Health Behaviour, Chronic Disease and Health Promotion). However, some points are listed below:

Page 1, line 5: please make name numbers superscripted.

Page 1, lines 6-13: please make affiliation numbers superscripted, and add space.

Abstract

Line 16: ‘physi-cal activity’. Delete The hyphen

Line 18: ‘We re-cruited’. Delete The hyphen

Line 24: ‘PA with higher BMI percen-tiles.’. Delete The hyphen

Line 25: ‘In the ad-justed models’. Delete The hyphen

Line 26: ‘with club sports participa-tion’. Delete The hyphen

1. Introduction

Line 79: ‘Therefore, this research study aimed to examine sociodemographic differences in perceived neighbourhood …’ this aim is not presented in the abstract.

Research hypotheses were not stated.

2. Materials and Methods

Line 88: ‘needy/needy areas’. What does it mean?

Line 142: ‘Moderate physical activity was described to participants as’. Needs a reference.

Line 143: ‘Vigorous physical activity was described as leading to harder’. Needs a reference.

Lines 145-49: ‘As overestimation is very common with self reported measurements of physical activity’. Need references.

Regarding to the different intensities (light, moderate and vigorous) used across the Physical Activity Questionnaire domains, you described both moderate and vigorous physical activity in lines 142-45, but you didn’t mention any description of light physical activity.

3. Results

No comments.

4. Discussion

Please start the aim of the study, and then a short sentence explaining the most important findings of the study like “The most important finding of the present study was…”. 

References

References are not adequate to IJERPH style whether within the text or in the reference section. For example:

Journal Articles:

1. Author 1, A.B.; Author 2, C.D. Title of the article. Abbreviated Journal Name (italic) Year (bold)Volume, page range.

Please check all references be compatible with IJERPH style (ACS style).

Best regards, 

Reviewer 3 Report

This paper presents a statistical analysis of the relationship of physical activity levels and BMI and perceptions of ‘walkability’ of South African neighbourhoods, using demographic comparisons. The methods and statistical tests appear appropriately selected and applied, although the convenience sample and slightly limited geographical spread makes generalisability of results more questionable. However, this limitation is recognised and acknowledged by the research team. The treatment of the individual existing empirical studies in the introduction is arguably a little cursory, however I am not particularly concerned by this and it does work for a paper that is clearly aiming for, and achieves, succinct and straightforward messaging (choosing to omit qualitative data at this stage). There is enough critique and rationale presented here to justify the research and, despite the methodological limitations noted above, I think the research makes a valuable contribution and is well-worthy of wider dissemination in its current form.
